# Association of Carboxyhemoglobin Levels with Peripheral Arterial Disease in Chronic Smokers Managed at Dr George Mukhari Academic Hospital

**DOI:** 10.3390/ijerph17155581

**Published:** 2020-08-02

**Authors:** Mashudu Nemukula, Motetelo Alfred Mogale, Honey Bridget Mkhondo, Lizette Bekker

**Affiliations:** 1Department of Biochemistry, School of Science and Technology, Sefako Makgatho Health Sciences University, Pretoria 0208, South Africa; alfred.mogale@smu.ac.za; 2Department of Pre-Clinical Sciences, School of Health Care Sciences, University of Limpopo, Polokwane 0727, South Africa; bridget.mkhondo@ul.ac.za; 3Department of Chemical Pathology, School of Medicine, Sefako Makgatho Health Sciences University, Pretoria 0208, South Africa; lizette.bekker@gmail.com

**Keywords:** chronic cigarette smokers, carboxyhemoglobin, nitric oxide, endothelial dysfunction, peripheral arterial disease

## Abstract

Chronic cigarette smokers (CCS) are known to have elevated levels of carboxyhemoglobin (COHb). However, it is not known whether increased levels of COHb are associated with endothelial dysfunction (ED), and therefore the development of peripheral arterial disease (PAD). The aim of the study was to investigate the association of blood COHb and plasma nitric oxide (NO) levels, and whether it is an independent risk factor in the development of PAD among CCS at Dr George Mukhari Academic Hospital (DGMAH). A sample of 120 CCS with PAD and a convenience sample of 100 CCS without PAD were recruited into the study. Blood COHb levels were measured using the ABL 90 FLEX CO-oximeter automated spectroscopy. Plasma nitric oxide (NO) levels were measure using ELISA. Logistic regression analysis was used to investigate the association of blood COHb and plasma NO with PAD. Blood COHb levels of CCS with PAD were significantly higher than those of CCS without PAD, and the NO levels of CCS with PAD were significantly lower than those of CCS without PAD. Although both the blood COHb and plasma NO in CCS were significantly associated with PAD in bivariate logistic analysis, only plasma NO was independently associated with PAD in multivariate logistic analysis. This finding is consistent with the hypothesis that COHb is a cause of arterial damage in PAD, leading to reduced NO, and therefore reduced arterial dilation.

## 1. Introduction

Research studies have established a strong association between cigarette smoking and atherosclerotic vascular diseases, such as peripheral arterial disease (PAD) and coronary artery disease [1,2,3]. PAD is characterized by the narrowing and blocking of the arteries supplying blood to the lower limbs. The most common clinical manifestation of PAD is critical limb ischemia, which is associated with a risk of limb loss. The pathophysiologic development of PAD results from any disease stimulating occlusion or stenosis of the lower limb arteries through atherosclerosis [4]. Available evidence also suggests that cigarette smoking, like most other atherosclerotic vascular risk factors, is known to promote the development of atherosclerotic vascular disease, through its negative effect on the endothelial function [5,6,7,8]. Endothelial dysfunction (ED) is generally defined as a failure of the endothelium to perform normal physiological functions including regulation of vascular tone, and anti-thrombotic, anti-inflammatory and anti-smooth muscle proliferation. Thus, ED is characterized by the tendency to increase vasoconstriction, pro-thrombic activity, vascular inflammation and vascular smooth muscle proliferation [9], as well as increased permeability to macromolecules and polymorphonuclear leukocytes [10]. Due to the fact that most of the physiological functions of the endothelium are mediated by nitric oxide (NO), ED is also defined by some authors as “reduced biosynthesis of NO” or reduced bioavailability of NO, is an established mediator of the atherosclerotic process [7,8]. A number of circulating biomarkers, including cotinine, a metabolite of nicotine, thiocyanate and carboxyhemoglobin (COHb), are used as indicators for chronic cigarette smoking [11,12,13,14]. Chronic cigarette smokers (CCS) are known to have elevated levels of COHb [15]. However, it is not known whether increased levels of COHb are associated with ED, and therefore the presence of PAD. Therefore, the aim of the study was to investigate whether blood COHb is correlated with plasma NO levels, and whether it is an independent risk factor of the development of PAD among CCS at Dr. George Mukhari Academic Hospital (DGMAH).

## 2. Materials and Methods

The study was a cross-sectional case-control study which investigated and compared the levels of blood COHb and plasma NO levels in CCS diagnosed with PAD, with those of CCS without PAD. The study was conducted at DGMAH, a tertiary hospital that serves as an educational site for the faculty of Health Sciences of Sefako Makgatho Health Sciences University (SMU) in Ga-Rankuwa, Pretoria, South Africa.

A non-probability sample of 120 CCS with PAD (case group), attending the surgical outpatient departmental (SOPD) clinic of DGMAH for medical review, and a convenient sample of 100 CCS without PAD (control group), were recruited into this study. The control group study participants were recruited mainly from staff members of SMU. The case group study participants were included in the study if they were chronic smokers of different ethnic groups, aged 18 years and above with clinical diagnosis of PAD, while the control group included chronic smokers of different ethnic groups, aged 18 years and above, without PAD or any other atherosclerotic macro vascular condition.

Both the case and control group study participants were excluded from the study if their occupation exposes them to a carbon monoxide (CO) polluted environment, such as mining, road work constructions and industrial exposure to CO, as well as those with severe anaemia (haemoglobin levels less than 8 g/dl). All study participants gave their informed consent after the purpose of the study and their rights were clearly explained to them. The study was conducted in accordance with the requirements of the research and ethics committee of the SMU (MREC/P/242/2014/PG). The sociodemographic characteristic such as age, gender, ethnic group and history of smoking (duration of smoking and average number of cigarette per day) were obtained by means of an unstructured data sheet form.

Venous blood samples for measurement of levels of COHb were collected from all study participants into blood collections tubes (BD Vacutainer^®^, Franklin Lakes, NJ, USA). In brief, 3 mL of whole blood were directly injected into the radiometer ABL 90 FLEX CO-oximeter to measure the levels of COHb. After the radiometer ABL 90 FLEX CO-oximeter had processed the whole blood samples, results were immediately printed.

For measurements of plasma NO concentration, venous blood samples were collected and measured using a Colorimetric Nitric Oxide Assay kit purchased from Sciencell Research Laboratories (Carlsbad, CA, USA). In brief, the measurement of NO concentration was estimated by quantification of nitrite (NO_2_), applying the Griess reaction according to the method of Green et al. [16]. During the reaction, nitrate (NO_3_) is converted to NO_2_ by using nitrate reductase and co-factor enzyme. Then NO_2_ reacts with sulfanilamide and heterocyclic amine of naphthylene-ethylene-diamine (Griess reagent) under low pH conditions, to form a chromophore with a characteristic absorption spectrum.

In brief, 285 µL of each sample was mixed with 15 µL of ZnSO_4_ (20 × strength) in a 1.5 mL micro tube. The mixtures were then vortexed for 1 min, centrifuged at 10,000 RCF for 10 min at 4 °C, and 100 µL/well of supernatant was transferred into each well of a 96-well plate. This was then followed by addition of a cocktail consisting of 100 µL of vanadium chloride and 50 µL each of Griess reagent I and Griess reagent II to each well of the 96-well plate. The reaction mixtures were incubated for 30 min at room temperature while protected from light. Thereafter, the absorbance of the mixtures was measured on Elisa Plate Reader (SpectraMax iD3 Multi-Mode Microplate Reader) supplied by Molecular Devices LLC. (San Jose, CA, USA) at the test wavelength of 540 nm and a reference wavelength of 630 nm, after which the 630 nm background absorbance was subtracted from the 540 nm measurement.

## 3. Data Analysis

All statistical calculations were performed using the Statistical Package for the Social Sciences (SPSS) software (Version 24), SPSS Inc., Chicago, IL, USA. Categorical data were expressed as mean ± standard deviation in tables. Comparison between different study groups was performed using the two sample *t*-test and the Wilcoxon rank sum for parametric and non-parametric data respectively. The statistical significance testing was set at the 0.05 (5%) level. Logistic regression analysis was performed to determine the association of blood COHb levels, NO plasma levels and selected sociodemographic characteristics with the presence of PAD. Both bivariate and multinomial logistic regression analysis were performed to determine whether COHb was an independent risk factor of ED among CCS. The significances level was set at *p* < 0.05.

## 4. Results and Discussion

Table 1 shows the sociodemographic characteristics of the study participants. As expected from common knowledge, there were more male than female subjects in both the CCS with PAD group and the CCS without PAD group. The average age of the CCS with PAD group, compared to those of the CCS without PAD group, was higher. There were more black people than white people in both the CCS with PAD group and CCS without PAD group. The fact that there were more black people than white people in both CCS with PAD and CCS without PAD was expected, since DGMAH has traditionally served, and is still serving, more black South Africans than white people. The duration of cigarette smoking for the CCS with PAD was higher, compared to the CCS without PAD. The CCS with PAD smoked more cigarettes per day compared to the CCS without PAD.

As shown in Table 2, the mean COHb level of the CCS with PAD group was found to be significantly higher than that of the CCS without PAD group (*p* < 0.0001). This observation could be partly explained by the fact that the duration of cigarette smoking and the number of cigarettes smoked per day were higher in the CCS with PAD group than those of CCS without PAD group. The measurement of COHb is widely used to monitor the smoking status of any individual. There is a direct relationship between the smoking habits of an individual and the levels of COHb [17]. One study found that the COHb levels in non-smokers range from 0.3% to 0.7%, rising to 3–8% particularly in smokers [18]. In our study, we found an averaged COHb level of 3.84% among the CCS with PAD group. Taking into consideration that COHb levels greater than 2% are used to separate chronic smokers and non-smokers [19], the observed levels of COHb in our study were still within the reference ranges for usual levels reported in cigarette smokers. However, Widdop et al. [20] showed that the COHb levels for intensive smokers range from 5% to 6%, implying that the observed COHb levels in our study are asymptomatic.

As shown in Table 2, it was observed in this study that the nitric oxide (NO) levels of the CCS with PAD group were significantly lower than those of the CCS without PAD group (*p* < 0.001). Since reduced levels of NO are an established marker for endothelial dysfunction, the findings of this study are in agreement with the notion that cigarette smoking, like other cardiovascular risk factors, promotes atherosclerotic vascular diseases, including PAD, through endothelial dysfunction [8,21,22,23,24].

Bivariate odds ratios were calculated in order to determine the associations between COHb levels, NO levels, gender and age of the study subjects, duration of smoking, and number of cigarettes smoked per day, with the development of PAD. The results shown in Table 3 indicate that PAD is more likely to present in patients with COHb blood levels greater than 2% [crude odd ratio (COR) = 4.42, 95% confidence interval (CI) (0.093–0.551) *p* value < 0.001], plasma NO levels greater than 24.6 µmol/L [COR = 6.65, 95% CI (0.068–0.339) *p* < 0.0001], duration of smoking more than 40 years [COR = 3.86 (0.120–0.763) *p* < 0.017] and an age of 60 years or above [COR = 3.89 (0.042–0.302) *p* value < 0.0001].

In bivariate analysis, COHb levels, NO levels, age above 60 years and duration of smoking were found to be associated with the presence of PAD among the study subjects. These findings were not unexpected, since all these factors are known risk factors for the development of atherosclerotic vascular diseases [25]. The fact that NO levels appeared to be independently associated with the development of PAD was expected, since this factor is an established predictor of atherosclerotic vascular diseases [26]. However, the observation that the main focus of the study, namely, blood COHb levels, was not independently associated with the development of PAD was an interesting surprise.

Multivariate logistic analysis was performed to identify independent risk factors of PAD among the study subjects. For the purpose of this analysis, determinants that were significantly associated with PAD (*p* < 0.05) in bivariate logistic regression analysis (COHb level, NO level, age > 60 years and duration of smoking > 40 years) were subjected to multivariate logistic regression analysis, using SPSS statistical package version 24. The results of this SPSS multiple logistic analysis are summarized in Table 4, and suggest that among the variables analyzed, only plasma NO levels were significantly and independently associated with the development of PAD among the study subjects.

## 5. Limitations and Recommendation of the Study

There are several limitations that should be taken into consideration when interpreting the results of the study. Firstly, the sample size was small, and it may impair our ability to separate signal from noise. Secondly, the study was a cross-sectional case-control study, and therefore, the cause and effect relationship could not be inferred from the study results. Thirdly, patients were recruited from a single hospital, rather than it being a community-based sample. Thus, the findings could not be generalized beyond these study samples. However, despite these limitations, the current study offers insights into the function of COHb levels and NO levels as potential risk markers for the development and progression of PAD among chronic cigarette smokers.

## 6. Conclusions

The result of the study confirms the notion that COHb levels are elevated in CCS with PAD, compared to their pears without PAD. The result of the study also suggest that NO levels are reduced in CCS with PAD, compared to their peers without PAD. Thus, although COHb levels do not appear to be independently associated with PAD, it can be hypothesized that COHb, like other known cardiovascular risk factors, may promote PAD through endothelial dysfunction. This finding is consistent with the hypothesis that COHb is a cause of arterial damage in PAD, leading to reduced NO, and therefore reduced arterial dilation. Further studies are needed in order to support or reject this hypothesis.

## Figures and Tables

**Table 1 ijerph-17-05581-t001:** Sociodemographic characteristics of the study subjects.

Variable	CCS with PAD	CCS without PAD
Gender:		
Male n (%)	108 (90)	83 (83.3)
Female n (%)	12 (10)	17 (16.6)
Average Age (yrs.)	56.6	47.5
Ethnic group:		
Black, n (%)	114 (95)	95 (95)
White, n (%)	6 (5)	5 (5)
DOS (Average Years)	30.2	27.78
NC/day	13.55	10.31

Notes: Data presented as mean for continuous variables and n (%) for categorical variables. CCS: chronic cigarette smoking; PAD: peripheral arterial disease; DOS: Duration of smoking; NC/day: average number of cigarettes per day.

**Table 2 ijerph-17-05581-t002:** Carboxyhemoglobin and Nitric oxide levels of the study participants.

Variables	CCS with PAD	CCS without PAD	*p*-Value
	(*n* = 120)	(*n* = 100)	
	Mean ± SD	Mean ± SD	
COHb	3.84 ± 1.32	2.93 ± 0.99	0.0001
NO	28.48 ± 9.86	41.75 ± 59.34	0.001

Note: Values are presented as mean ± standard deviation (SD), COHb, Carboxyhaemoglobin; NO, Nitric oxide; CCS, Chronic smokers; PAD, Peripheral arterial disease.

**Table 3 ijerph-17-05581-t003:** Bivariate logistic regression analysis of the association between both COHb and NO with the presence of PAD.

Factor	Without PAD *n* (%)	With PAD *n* (%)	COR, 95% CI	*p*-Value
**[COHb]**				
<2%	30 (25)	7 (7)	1.0 (ref)	
>2%	90 (75)	93 (93)	4.42 (0.093–0.551)	<0.001
**[NO]**				
<24.6 µmol/L	92 (76.6)	33 (33)	1.0 (ref)	
>24.6 µmol/L	28 (23.4)	67 (67)	6.65 (0.068–0.339)	<0.0001
**Gender**				
Male	108 (90)	83 (83)	1.0 (ref)	
Female	12 (10)	17 (17)	1.84 (0.235–1.252)	<0.426
**Age**				
≤40 (yrs)	32 (26.7)	10 (10)	1.0 (ref)	
40–60 (yrs)	74 (61.7)	52 (52)	2.25 (0.197–1.01)	0.055
≥60 (yrs)	14 (11.6)	38 (38)	3.89 (0.042–0.302)	<0.0001
**NC/day**				
≤10	74 (61.7)	42 (42)	1.0 (ref)	
10–19	32 (26.7)	33 (33)	1.81 (0.283–1.025)	0.163
≥19	14 (11.6)	25 (25)	1.74 (0.140–0.696)	1
**DOS**				
≤20	40 (33.3)	30 (30)	1.0 (ref)	
20–40	70 (58.3)	45 (45)	0.86 (0.623–2.187)	1
≥40	10 (8.4)	25 (25)	3.86 (0.120–0.763)	0.017

Notes: PAD, peripheral arterial disease; COHb, carboxyhemoglobin; NO, nitric oxide; NC/day, average number of cigarettes per day; DOS, duration of smoking.

**Table 4 ijerph-17-05581-t004:** Multivariate logistic regression analysis of factors that were associated with PAD during bivariate analysis.

PAD ^a^	B	Std. Error	Wald	df	Sig.	Exp (B)	95% Confidence Interval for Exp (B)	
							Lower Bound	Upper Bound
Intercept	−1.968	0.683	8.292	1	0.004			
[COHb = 0]	1.207	0.683	3.121	1	0.077	3.345	0.876	12.766
[COHb = 1]	0 ^b^			0				
[NO = 0]	1.417	0.431	10.818	1	0.001	4.124	1.773	9.593
[NO = 1]	0 ^b^			0				
[AGE = 0]	2.101	0.921	5.204	1	0.023	8.174	1.344	49.700
[AGE = 1]	1.143	0.631	3.277	1	0.070	3.136	0.910	10.813
[AGE = 2]	0 ^b^			0				
[DOS = 0]	−0.870	0.973	0.800	1	0.371	0.419	0.062	2.819
[DOS = 1]	0.435	0.792	0.302	1	0.583	1.546	0.327	7.302
[DOS = 2]	0 ^b^			0				

^a^ The reference category is: PAD. ^b^ This parameter is set to zero because it is redundant. Notes: PAD, peripheral arterial; NO, nitric oxide; DOS, duration of smoking.

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
