# Peer review of "Association of Carboxyhemoglobin Levels with Peripheral Arterial Disease in Chronic Smokers Managed at Dr George Mukhari Academic Hospital"

_ijerph, 2020, doi:10.3390/ijerph17155581_

Round 1
Reviewer 1 Report
Dear Authors,
The manuscript ID: ijerph-752027 entitled “Association Of Carboxyhemoglobin Levels With Peripheral Arterial Disease In Chronic Smokers Managed At Dr George Mukhari Academic Hospital” has been corrected.
The answers to the comments of the reviewer are satisfactory. The changes introduced by the Authors improved the quality of the article and the clarity of the information provided. However, the Authors should make some minor editing corrections, for example:
- ml should be mL,
- in Table 4 the values written as ".683" should be 0.683 and so on.
Author Response
- ml was changed to mL following the reviewers recommendations.
- The values written as .683 were written as 0.683 and so on.
Reviewer 2 Report
Thanks for a good rebuttal, the paper is much improved.
This is a very interesting approach to better understand what in cigarette smokers might differentiate those who have PAD vs those who do not have PAD.
There are still small language errors; please proof read for grammar.
L 17: “One hundred and twenty CCS with PAD and 100 CCS without PAD study participants were conveniently recruited into the study.” Better wording: “A sample of 120 CCS with PAD and a convenience sample of 100 CCS without PAD were recruited into the study.”
L 23: “Although both blood COHb and plasma NO were significantly associated with PAD in bivariate logistic analysis, only plasma NO was independently associated with PAD in multivariate logistic analysis.” For clarity, I would have added “in CCS”, namely “Although both blood COHb and plasma NO in CCS were significantly associated with PAD in bivariate logistic analysis, only plasma NO was independently associated with PAD in multivariate logistic analysis.” The abstract doesn’t have an interpretive conclusion. My interpretation would be the following: “This finding is consistent with the hypothesis that COHb is a cause of arterial damage in PAD, leading to reduced NO and therefore reduced arterial dilation.” I would suggest adding this type of interpretation to the Discussion, also. Table 4 shows p = 0.077 in the joint model for COHb; I would not be surprised if COHb had effects beyond the intimal damage, but apparently a larger sample size is required to unequivocally rule out chance in the COHb association.
L 80-86: I looked at the ScienCell Research documentation (https://www.sciencellonline.com/colorimetric-nitric-oxide-assay.html). I accept the assay, which I had not previously heard about. I would quote from their statement: “NO is extremely unstable and undergoes rapid oxidative degradation to nitrite (NO2-) and nitrate (NO3-), which can be spectrophotometrically determined. ScienCell's Colorimetric Nitric Oxide Assay kit provides an accurate measurement of NO level in a simple two-step process:” I corrected a typo (repaid to rapid). Your statement also did not mention NO3-; it should mention that. Note the odd spelling of the company name.
L 123: CSS should be CCS.
L 131: “However, Widdop et al. [20] showed that COHb levels for intensive smokers ranges from 5%-6%, implying that the observed COHb levels in our study are not asymptomatic.” What does “asymptomatic” mean in this sentence?
L 142-148: You have created odds ratio for less likely for non-PAD to be high, rather than more likely for PAD to be high. Please invert the odds ratios to be consistent with your text. For example, for COHb, 93*25/(7*75) = 4.42.
Table 4: I don’t know what “no PAD” means on the left side of the table. I think it can be deleted.
Author Response
Please see the attachment

This manuscript is a resubmission of an earlier submission. The following is a list of the peer review reports and author responses from that submission.
Round 1
Reviewer 1 Report
The manuscript ID: ijerph-752027 entitled “Association Of Carboxyhemoglobin Levels With Peripheral Arterial Disease In Chronic Smokers Managed At Dr George Mukhari Academic Hospital” is an attempt to find a relationship between carboxyhemoglobin levels and PAD in smokers.
In our knowledge about the effects of smoking on cardiovascular diseases, there are still many unexplained problems, so such a study, although limited to participants from one hospital, may be of interest.
The following points have to be refined by the authors:
Major Comment:
The Authors should change the discussion text or the layout of sections at work. Sections 4 (Results) and 5 (Discussion) should be combined into one point 4. Results and Discussion.
It is unacceptable to repeat exactly the same sentences as e.g. line 111-119 and 166-174 or 124-129 and 175-180.
Similarly, from section 6, delete the sentence from line 197-200 (Although COHb levels do not appear to be independently associated with PAD, it can be hypothesised that COHb like other known cardiovascular risk factors may promote PAD through endothelial dysfunction. Further studies are needed in order to support or reject this hypothesis), which is exactly repeated in 7, lines 204-207.
Minor comments:
- the literature cited in the paper comes mainly from 2000-2010 and is even older. Have there been no reports in this area in the last 10 years?
- Line 43 - undoubtedly the sentence is true, but did the authors find the full publication on this subject since they only cited the abstract? [14] - Light, A.; Grass, C.; Pursley, D.; Krause, J. Carboxyhemoglobin levels in smokers vs. non-smokers in a smoking environment. Respir Care. 2007; 52(11):1576. 2007 OPEN FORUM Abstracts
- Line 72 - Is it known how long after the last cigarette was smoked, blood samples were taken, and how this could affect the carboxyhemoglobin level?
- Line 115 – “However, the average COHb levels of 3.84 % observed among the CCS with PAD group was still within the reference range reported for the normal levels in COHb in cigarette smokers [15]” - as you can read in [15], the range for intensive smokers: "5- 6% Normal levels in tobacco smokers",
- Line 75 – “3mls” is this an abbreviation for the official SI unit?
- Line 174 – [22] - does not contain COHb level information - the references should be ordered.
The abbreviations used by the authors should be explained in the place where they appear for the first time, for example: P? and p?, COR, CI.
There are also typographical errors in the work, e.g. the information contained in the tables are located in different lines which makes it difficult to fit them - the tables should be ordered.
References - the year of publication should be written in bold. Ref [7] - missing year of publication and doi number. The Authors do not use abbreviations of journals titles and in many places the doi number is missing.
Reviewer 2 Report
The article has numerous language errors.
Because NO is an ephemeral molecule, you should add something about how the assays measures it.
Nothing about disease history is stated, apart from a vague exclusion criterion for the non PAD sample.
L 54: The sampling scheme assures that the control group is substantially healthier than the case group. This is a limitation that may affect validity of the findings.
L 71: “structured data sheet” Does this mean that these characteristics were self-reported by the participant without assistance of an interviewer?
Was the blood sample venous?
The nonsmoker control group should be deleted; the age difference is so large that findings for it defy interpretation in the context of the PAD and nonPAD smokers.
L 104: You do not have an experimental group, rather you have a PAD case group. Experimental means you are doing something to them, not under their control.
L 104: “The duration of cigarette smoking of the experimental group was higher compared to the control group (chronic smokers without PAD).“ Because the PAD cases were on average 8 yrs older than the non PAD control, the added duration of 3 years actually suggests less lifetime smoking in the cases than in the controls.
L 117: The phrase “normal levels of COHb in smokers” is more than odd. Smoking is not normal. Perhaps you mean “usual levels”, but the levels can still be remarkable. Anyway, this finding is not helpful, the comparison earlier in the paragraph that the COHb level is higher than in the nonPAD controls is what is relevant.
Figure 1 and 2: Age is a strong confounder and should be adjusted for. Figures 1 and 2 would be more informative if they were a table; the main thing they do is give the unadjusted distributions, so tabulated mean and standard deviation would be helpful.
L 136: “PAD is more likely to develop” Your design does not address development, you can say “is more likely to be present”
L 137: What does COR mean? Some sort of odds ratio?
Table 2: Why did you dichotomize COHb and NO? The findings in Table 3 might be different for different parts of these distributions.
L 191: “Firstly, the sample size was small, therefore making it difficult to generalize the findings.” Small sample size may impair ability to separate signal from noise, but is not related to generalizability.